# Sexual Dimorphisms in Neurodevelopment May Affect TBI Recovery in Pediatric Patients

**DOI:** 10.3390/biomedicines13123033

**Published:** 2025-12-10

**Authors:** Moira F. Taber, Franklin D. West, Erin E. Kaiser

**Affiliations:** 1Regenerative Bioscience Center, University of Georgia, Athens, GA 30602, USA; 2Department of Animal and Dairy Science, College of Agricultural and Environmental Sciences, University of Georgia, Athens, GA 30602, USA; 3Neuroscience Program, Biomedical and Health Sciences Institute, University of Georgia, Athens, GA 30602, USA

**Keywords:** traumatic brain injury, pediatric, sex, neuroanatomy, neurodevelopment, white matter, gray matter, connectivity

## Abstract

Traumatic brain injury (TBI) is a leading cause of death and disability, with broad heterogeneity in recovery outcomes particularly noted in pediatric patients. Children post-TBI are vulnerable to aberrant neurodevelopment, specifically in structural and functional neural networks as they correlate with cognitive, behavioral, and motor function outcomes. Consideration for sex as a biological variable which innately influences neuroanatomy, neurodevelopment, and functional organization may elucidate risk factors for negative outcomes in pediatric TBI. For example, TBI damage in sexually dimorphic neural structures and networks may explain deficits in social cognition, working memory, as well as internalizing and externalizing behaviors, which differentially impact the quality of life in male versus female TBI patients. However, characterization of sex in conjunction with developmental patterns in normal and injured pediatric populations is limited due to small sample sizes, the low representation of females, a lack of longitudinal data, and the utilization of analyses that are not sensitive enough to detect subtle differences in TBI pathologies and recovery between the sexes. This review aims to analyze and synthesize the existing evidence regarding the influence of sex on the developmental trajectories of neuroanatomical structures, white and gray matter compartments, and the network disruptions that align with sex-specific functional recovery outcomes following pediatric TBI. The delineation of these sex influences will facilitate better precision-based medicine approaches to improve patient outcomes.

## 1. Introduction

Pediatric traumatic brain injury (TBI) is well established as a leading cause of mortality and disability, causing an immense public health burden with >500,000 emergency department (ED) visits and 2000 deaths annually in the United States [1,2]. Despite the declining incidence of severe pediatric TBI, likely due to improved safety and education measures, TBI-related ED visits have continued to rise, due in part to the increasing incidence and reporting of mild pediatric TBI cases [1]. Pediatric risk for TBI is of particular concern due to coalignment with critical neurodevelopmental periods and disruption of brain development (e.g., delayed brain growth and abnormalities in white matter microstructure) which leads to negative functional outcomes (e.g., learning, memory, and motor function) [3]. Given the heterogeneity in TBI pathophysiology and recovery, research efforts have focused on patient- and injury-specific delineation of risk factors to coordinate precision-based medical approaches and potential therapeutic targets. Sex has recently been identified as a key biological variable in TBI pathophysiology and a mediator of recovery outcomes.

Human and preclinical TBI studies have shown sex-related differences in cognitive, behavioral, and motor dysfunction, in addition to somatic symptom presentation and persistence in adults [4,5,6]. The neuroprotective effects of estrogen and progesterone may partially explain these sex-dependent recovery outcomes, but studies have been unable to draw consistent conclusions, as therapeutic administration of sex hormones have failed in clinical trials [7]. Additionally, hormonal profiles fail to account for observed sex effects in pediatric populations where sex hormone production is limited, thus suggesting the role of sex to be more complex than previously determined.

Age is well defined as an important predictor for functional recovery post-TBI [8,9]. Age and sex interactions further influence TBI recovery in children as sex differences in developmental patterns can influence individuals’ vulnerability to chronic perturbations in response to damage. The consideration of sex in pediatric TBI pathology and recovery has been detrimentally understudied, often due to low sample sizes and underrepresented patient populations. Convergent evidence suggests that the delineation of sex effects could have important implications for TBI pathophysiology and treatment development. This review aims to analyze and synthesize the existing evidence regarding the influence of sex on the developmental trajectories of neuroanatomical structures, white and gray matter compartments, and the network disruptions that align with sex-specific functional recovery outcomes following pediatric TBI, thereby highlighting how these factors can inform and enhance precision medicine approaches in future research endeavors.

### Search Strategy and Selection Criteria

The articles in this review were retrieved using the following electronic databases: PubMed, MEDLINE, Google Scholar, and Web of Science. The search was limited to articles published between 1986 and 2025. Search terms were identified in the title, abstract, and key words using the following search terms: traumatic brain injury, TBI, pediatric/juvenile/children, sex, gender, gray matter, white matter, basal ganglia, limbic system, cognition, behavior, motor function, social cognition, attention, memory, working memory, atrophy, neurodegeneration, alcohol, putamen, globus pallidus/pallidum, hypothalamus, pituitary gland, thalamus, hippocampus, cerebellum, diffuse axonal injury, structural connectivity, diffusion tensor imaging, microstructure, cortical thickness, total gray matter volume, gray matter density, symmetry, functional connectivity, intrinsic connectivity, extrinsic connectivity, resting-state functional magnetic resonance imaging/rs-fMRI, task-based functional magnetic resonance imaging/tb-fMRI, default mode network/DMN, sensorimotor network, and attentional network.

## 2. Sexual Dimorphisms in Specific Brain Structures

Recent evidence suggests that fundamental sex differences in brain anatomy and neurodevelopment contribute to the observed differences in TBI severity and in the consequent prognosis between male and female children [10,11,12,13]. Even when controlling for the larger brain volumes of males, these sex differences are maintained in multiple gray and white matter structures well into adulthood [14,15,16]. Sex differences in neuroanatomy and development arise during fetal development through the masculinizing effects of gonadal and local steroid hormones on the bipotential embryonic brain, while working alongside sex chromosome-specific transcriptional profiles and microglia-mediated modulation [17,18]. Specifically, sex effects within the basal ganglia, limbic system, and cerebellum, among other brain structures, influence the complex nature of functional outcomes post-TBI (Figure 1). Considering the heterogeneity of TBI recovery, fundamental sex differences in cortical and subcortical structure size and development may influence patient recovery more than previously thought and thus remain critical topics of intensive discussion.

### 2.1. The Basal Ganglia

The basal ganglia consist of a group of connected subcortical structures located at the base of the forebrain. These structures play critical roles in decision making, reward-based learning, motor function, and addiction, and contribute to unique and specialized deficits following damage from neural injury. Sex-specific contributions to cellular signaling and proliferation underlie volumetric differences. However, the sex-specific role in these areas is still under active investigation. Female rat striata demonstrate higher densities of both dopaminergic fibers and GABAergic neurons throughout prenatal development [19]. Furthermore, fetal androgen-related masculinization of the brain indirectly induces microglial phagocytosis of newborn astrocytes for reduced density in the amygdala, which predilects rougher play in male rats [18]. In 2024, Backhausen et al. found significantly larger volumes in 14–24-year-old males versus females across subcortical regions including the caudate nucleus, putamen, globus pallidus, amygdala, and nucleus accumbens [20]. This research expanded upon previous studies in which sex-specific structural differences were observed in the putamen and globus pallidus in children and adults [15,16]. Interestingly, sex differences in age-related developmental trajectories were also observed in these basal ganglia structures, with normal healthy adult males demonstrating a higher degree of progressive atrophy in basal ganglia structures as they age relative to females [14,21]. Basal ganglia volumes peak in mid-adolescence with the maximum striatal volume observed at 10-12-, the maximum pallidum volume observed at 17, and the maximum amygdala volume observed at 19 years of age [22,23]. TBI-induced tissue atrophy in the basal ganglia in youths prematurely reduces this natural age-related trajectory, thus resulting in disproportionately greater cognitive and emotional deficits in males as they progress into adulthood [20,24,25,26,27].

In addition to differing psychopathology between males and females, TBI-induced damage to basal ganglia structures has been linked to increased addiction-related behaviors [28,29]. For example, seminal animal studies have identified sex differences in cellular mechanisms that are independent of pubertal hormones, such as increased D1 and D2 dopamine receptor density, and pruning in male mice as well as the estradiol-driven regulation of dopamine release and binding in female mice, which ultimately shape reward processing and reinforcement learning, core processes implicated in drug addiction [30,31,32,33,34]. Recent human functional MRI (fMRI) investigations supported these findings, as reward prediction error-related brain activity was enhanced in women compared to men and to an even greater extent when estradiol levels were elevated in both sexes. This reflects the influence of estrogen on the mesocorticolimbic dopaminergic pathway as it traverses the basal ganglia [35,36]. Because substance use disorders and alcohol addictions are common long-term outcomes post-TBI, researchers believe this may be due to tissue damage within the basal ganglia. One study found that neural activity in the fronto-basal ganglia network of TBI patients aged 9–10 years old predicted increased odds of sipping alcohol by ages 11–13, with males reporting significantly greater alcohol sipping than females [37]. In a study of students enrolled in grades 7–12, Ilie and colleagues found that males with a history of TBI were also more likely to report daily nicotine use than females [28].

It is important to note that future studies evaluating post-TBI addiction should also consider alternative addiction-related behaviors such as internet addiction, social media addiction, gaming disorders, and mobile phone addiction, all of which are more accessible to children and more socially acceptable for females. For example, an assessment of childhood TBI patients, aged 0–17 years old, found that males and females displayed significant differences in internalizing versus externalizing behaviors, such that males were more likely to report substance abuse and females were more likely to report anxiety and depression [38]. Young females with depressive symptoms had significantly higher risks of internet addiction relative to males [39], while also demonstrating a higher risk of mobile phone addiction [40,41]. Sex differences in psychosocial norms and symptom reporting influence the documentation of addiction disorders in TBI patient populations.

#### 2.1.1. The Putamen

The putamen is involved in everyday functions, including speech and language, reward processing, cognition, and motor control. During normal brain development, the putamen has both age-related volume decreases as well as sex-related volumetric differences, with larger putamen volumes noted in male children beginning around 8 years of age [42,43,44,45]. Females and males also have different putamen development trajectories, where putamen volumes in female children decrease until approximately 20 years of age. After 20 years old, the rate of volumetric atrophy decreases in females. Comparatively, putamen volumes in male children have a slower decrease until approximately 16 years of age. After 16 years old, the rate of volumetric atrophy increases in males [20,46,47]. In moderate- to severe-TBI patients, volume loss in the putamen contributes to deficits in cognitive function, attention, emotional control, motor function, and behavior [25,48,49]. In addition to volumetric differences, differences in dopamine signaling via the activation of D2 receptors between males and females may also contribute to differences in motor function deficits post-TBI [50,51]. Specifically, healthy females have demonstrated a 7–8% higher D2 receptor expression in the putamen and, when coupled with differences in putamen volumes, these findings reveal a novel mechanism for stunted motor function recovery in males post-TBI [14,52,53,54,55].

Putamen volume may also serve as a structural neural marker for future comorbid neuropsychiatric and neurodegenerative conditions [56]. Comparable neurological studies have found that putamen volume is significantly reduced in patients diagnosed with Huntington’s disease [57], dementia with Lewy bodies [58], Alzheimer’s disease [59], multiple sclerosis [60], attention deficit hyperactivity disorder (ADHD) [61], and major depression [62]. In a study investigating subcortical gray matter volume loss and its relation to executive dysfunction following TBI, the loss of putamen volume was significantly associated with poorer task-switching performance [63]. Lower functional independence scores were also found to be correlated with putamen atrophy [49]. In a 5-year longitudinal study, Simeone and colleagues found that TBI induced progressive volumetric atrophy in subcortical gray matter structures, including the putamen, to occur at a similar rate to aggressive neurodegenerative diseases, and was associated with lowered episodic memory and executive function compared to healthy controls [64]. In mice, males were shown to be more susceptible to neurodegeneration following TBI [65]. Jenkins et al. reported distinct striatal dopamine abnormalities in TBI, providing evidence that disrupted putaminal dopamine signaling enhances TBI deficits [51]. Collectively, these studies provide evidence a patient’s biological sex, age, and putamen volume constitute as risk factors that predispose individuals, particularly males, to worse outcomes post-TBI.

#### 2.1.2. The Globus Pallidus

The globus pallidus, also known as the pallidum when considering both the internus and externus halves, controls conscious and proprioceptive movement, and projects into all the other basal ganglia structures. Since the globus pallidus is a critical structure in basal ganglia circuits, characterization of its developmental trajectory is essential to gaining a better understanding of motor and non-motor dysfunctions as a consequence of TBI. This gray matter structure possesses both age- and sex-dependent volumetric differences. Males have a greater increase in volume until 16 years of age, followed by a sharp decline later in life [14,20]. This contrasts with female neurodevelopment, as the globus pallidus volume has been found to steadily increase until 24 years of age [14,20]. Understanding how these trajectories influence motor behavior deficits following TBI has become a major research objective, as recent evidence suggests that volume loss in the globus pallidus due to diffuse injury from TBI is significantly correlated with functional deficits [66,67].

Although evidence linking the globus pallidus to TBI outcomes comes primarily from adult studies, analyzing these recent findings is essential to guide and inform future pediatric research. Gooijers et al. demonstrated an association in adults between reduced globus pallidus volumes and worse bimanual behavioral deficits as determined by alternate finger tapping performance in the TBI group [53,68]. United States military service members with a history of mild TBI demonstrated an association between significantly smaller globus pallidus volumes and reductions in processing speed performance involving motor output when compared to controls [69]. These studies demonstrate globus pallidus volumes serve as a valuable prognostic marker of chronic neural pathology, which complicates motor control rehabilitation in TBI patients. Interestingly, Niemann et al. demonstrated in older adults that motor fitness positively correlated with the volume of the globus pallidus and that coordination training increased globus pallidus volumes, leading to improved patient recovery [70]. This suggests that the globus pallidus may serve as a novel therapeutic target for more individualized rehabilitation programs and/or treatment interventions such as deep brain stimulation. Given that globus pallidus volumes influence TBI recovery in adults, researchers should investigate its role in pediatric TBI wherein differing neurodevelopmental trajectories, such as the greater prepubertal volume in males, may confer differential protection against motor deficits.

### 2.2. The Limbic System

The limbic system is also susceptible to damage following TBI, which can result in chronic negative recovery outcomes in cognition, behavior, and autonomic nervous system regulation [25,71]. Sexually divergent limbic structures include the hypothalamus, thalamus, and hippocampus. The hypothalamic-pituitary axis is also critically important for the endocrine system, which influences many aspects of neurodevelopment including homeostatic function. Cellular sex differences in limbic structures include differences in the timing of cell maturation in the hypothalamus preoptic area, mast cell-driven neuronal excitation in thalamic neurons in female rats, and greater microglial phagocytic activity in the neonatal female rat hippocampus [72,73,74]. These cellular sex differences show that limbic regions develop within distinct neuroimmune, glial, and excitatory environments, demonstrating that the limbic system is shaped by fundamentally different cellular processes in males and females.

#### 2.2.1. The Hypothalamus and Pituitary Gland

Recent studies involving the limbic system have revealed MRI-based differences in male and female hypothalamic and pituitary gland volumes, with 7–9-year-old males exhibiting significantly larger hypothalamus and pituitary gland volumes compared to age-matched females [16,75,76,77]. Interestingly, this trend is reversed as 14–17-year-old females possess significantly larger pituitary gland volumes compared to age-matched males [77]. Hypothalamic development rates also differ between males and females, with males reaching maximum hypothalamus volumes at age 18 and females reaching maximum hypothalamus volumes earlier, at 15–16-years-old, due to differences in cell maturation timing [72,78]. These differences are critically important when considering that the hypothalamus plays a key role in maintaining homeostasis and regulating the endocrine system through hormonal signaling to the pituitary gland. Furthermore, the structural vulnerability of both the hypothalamus and pituitary gland, along with sex-specific differences in brain volume and developmental trajectories, provide a therapeutic target for the diverse hormonal deficiencies observed in TBI patients.

A comprehensive review of the literature spanning 22 years found TBI-related hormonal deficits ranged from 22.5 to 86% and multiple hormonal deficits ranged from 5.9 to 50% in the studied pediatric population [79]. Additional studies found hypopituitarism incidence ranged from 5 to 70% in TBI patients, making hypopituitarism one of the most prominent comorbidities [80,81]. Hypopituitarism has been found to delay puberty in pediatric TBI populations and can cause physical and psychological dysfunctions that last well into adulthood [25,82,83]. Additional studies by Gray and Prodam et al. go on to report a clear association between hypopituitarism and adverse cognitive outcomes such as memory, attention, language, physical conditioning, and mood disorders post-TBI [84,85]. In addition to hypopituitarism outcomes, specific growth hormones and gonadotropins, which are produced by the pituitary gland and regulate sexual development, reproduction, and growth, are commonly deficient in TBI patients [25,80,86]. Acerini and Lee et al. have identified these deficiencies as the primary reason for the observed differences in TBI recovery outcomes between male and female children [87,88]. Niederland et al. reported that 42% of all TBI children showed insufficient growth hormone response and 73% of these cases were males [89]. Agha et al. determined that decreased sex hormones affected primarily males post-TBI, with lasting effects on fertility, psychosexual function, and general well-being [90]. It was also found to affect their level of energy and motivational status, and the mortality rate of male patients secondary to cardiovascular disease [90]. The intricate relationship between hypothalamic and pituitary gland volumes and development in collaboration with TBI location and severity are important considerations to identify potential endocrine complications, which could be life-threatening and impact patients’ overall quality of life.

#### 2.2.2. The Thalamus

The thalamus primarily functions to receive, process, and relay sensory information to the cerebral cortex, thereby contributing to states of consciousness, attention, cognition, and memory. MRI-based studies report sex differences in thalamic volumes and developmental trajectories from childhood through adulthood [14,15,16]. Discrepancies in the literature regarding which sex has greater thalamic volumes is due to differences in sample size and age ranges. These findings collectively warrant further rigorous testing in preclinical studies as well as improvements in defined patient cohorts in clinical datasets. For example, Xie et al. and Sowell et al. demonstrated that sex affected thalamus size, with female children and adolescents having larger thalami relative to gray matter and brain volumes [43,91]. More recently, Wagner and colleagues saw lateralization effects in pediatric populations where females’ left thalamus volume increased by 17.3% and peaked at 16–17 years old, and their right thalamus volume increased by 11.4% and peaked at 11–12 years old [78]. Comparatively in males, the left thalamus volume increased by 23.2% and peaked at 15–16 years old, and the right thalamus volume increased by 20.7% and peaked at 17–18 years old [78]. This illustrates the delicate balance between sex and age with regard to thalamic development. Given these differences, it is important to consider that the impact of TBI on the thalamus may vary depending on the child’s age at the time of injury. The thalamus is particularly vulnerable to shearing effects during TBI, and consequent interruptions to thalamic development and volume loss have been shown to reduce cognitive and motor performance [25,53].

Reduced cortical thickness and deformation in key thalamic subregions that operate as part of the dorsolateral prefrontal cortex circuitry were significantly correlated with decreased executive functioning in pediatric patients [92]. Although not directly examined, understanding the role of the thalamus in executive functions such as planning and emotional regulation adds nuance to findings reported by Keenan and colleagues, wherein males aged 6–11 years old exhibited poorer executive function outcomes post-TBI [93]. One outcome of note was emotional control with additional evidence suggesting that male children demonstrated greater deficits in planning and organizational tests after injury [93]. Another study found male children were predisposed to new onset compulsions which were significantly associated with thalamic lesions induced by TBI [94]. This evidence indicates that thalamic integrity is a key driver of sex-specific TBI recovery, as it corresponds with male-predominant deficits, including impaired executive control of compulsions, emotion, and planning. Together these findings solidify the thalamus as a critical locus of divergence in male versus female functional outcomes.

#### 2.2.3. The Hippocampus

The hippocampus is an integral structure of the limbic system and is primarily associated with declarative and spatial memory. More specifically, the hippocampus is responsible for autobiographical memory and conversion of short-term memory to long-term memory [95]. During pediatric development, the hippocampus undergoes increased growth trajectories from 8 to 10 years old and peaks in volume at approximately 16 years old [78]. Sex differences have been widely reported, with some researchers reporting that female children possess greater hippocampal volumes, which are maintained throughout adolescence [16,95], while others report larger volumes in males [23]. It is important to note that in order to avoid such discrepancies, future investigations should not only account for differences in total brain volumes between males and females, but also avoid voxel-based morphometry technologies which have been found to be too ambiguous to consider sex-based volumetric differences and age trajectories [16,23,95]. With these analysis methods considered, the hippocampus was found to be significantly smaller in males compared to females at 8–15 years old when controlling for both total brain volume and body size [96,97,98].

Sex-related differences in hippocampal development trajectories bear conflicting evidence. One study reported that smaller hippocampi volumes in males were accompanied by decreased hippocampal development rates when compared to females [99]. Longitudinal studies have not found sex-related differences in hippocampal developmental trajectories [100,101]. Meanwhile, observational neuroimaging data suggest that differences in hippocampal development is sensitive to sex steroid and chromosome effects [98]. In a sample of 8–15-year-old males and females, Neufang et al. found that a larger hippocampus volume was related to increased levels of testosterone in both sexes [98,102,103]. Peper et al. showed changes in sex steroid availability during puberty and adolescence triggered structural reorganization of the hippocampus in the developing brain, as early pubertal females had less gray matter density in the hippocampus compared to non-pubertal females [102,103,104]. Collectively, variations in hippocampal segmentation methods contribute to observed differences in developmental trajectories [105] and it is thereby recommended that conclusions regarding sex biases in volumetrics and the timing of hippocampal maturation must include functional data analysis tools (e.g., multi-atlas automated subcortical segmentation and surface area change trajectory models) for multiple brain regions [106].

TBI-induced tissue damage to this critical brain region also demonstrates sex bias. Inherent volumetric differences between male and female children results in differing hippocampal function and recovery. A loss of >20–30% of the dorsal hippocampus and >39–52% of the ventral hippocampus is required to observe hippocampal-dependent learning impairments [107]. Inherently smaller hippocampal volumes in male children underlies greater susceptibility to cognitive deficits, including more pronounced impairments in learning and declarative memory compared with females [95,108,109]. Meanwhile, increased hippocampi volumes in females accounts for better performance in verbal and non-verbal memory assessments post-TBI compared to those in male children [110]. Bird and Farace et al. found that inherently larger hippocampi volumes in females contributed to the improved encoding of long-term memories as well as spatial processing and navigation [111,112]. In the context of hippocampal involvement in the social brain network, male mice demonstrated greater impairments in terms of psychosocial behavior, particularly in social recognition, following pediatric TBI [113]. These deficits have been linked to alterations in neuronal morphology within the hippocampus and medial prefrontal cortex [113]. Davila-Valencia and colleagues further found neurogenesis within the hippocampus to differ between males and females, with females showing reduced proliferation and short-term survival of newborn cells, but ameliorative effects were present in males. These results correlated with males exhibiting less depressive behavior [114]. Collectively, these studies highlight the need to consider post-TBI hippocampal damage within the context of developmental rates, and atrophy trajectories, as it differentially impacts learning and memory capabilities between male and female children.

### 2.3. The Cerebellum

The cerebellum is responsible for a diverse number of functions ranging from motor control to cognition and emotion. Highly connected to the rest of the brain, the cerebellum influences specific aspects of cognitive development, including executive function, cognitive planning, personality, and learning [115,116,117]. Cellular mechanisms underlying cerebellar sex differences include chromosomal regulation of Calbindin (Calb) D28K mRNA expression, where the XX chromosome corresponded to increased Calb levels in mice [118]. On a larger scale, recent evidence has found that cerebellar gray matter is closely related to cognitive function and volumetric analysis, making it a reliable predictor of cognitive performance in children [116]. Furthermore, both sex and age have been identified as key mediators of cerebellar development [115,116,117]. Although male children demonstrated greater absolute gray matter volumes than females, distinct cerebellar lobules have been found to be larger in females [15,115,119]. Female cerebellar volume has been found to peak around 12 years of age, as opposed to males who peak closer to 16 years of age, thus providing further evidence of the interplay between age and sex [23,115]. Additionally, the cerebellum develops in an anterior-to-posterior gradient in 6–10-year-old children, where the posterior gray matter volume is larger than that of the anterior [117]. This white–gray matter ratio adapts from a negative to a positive relationship with cognitive functioning in adolescence, and thus is indicative of cognitive function acquisition during childhood [115,117]. Together, these findings underscore the dynamically regulated role of the cerebellum, as its cellular mechanisms and structure actively contribute to the age- and sex-dependent trajectories of childhood brain maturation.

Evidence for sex-based differences in cerebellum-related functional recovery following TBI exists at the genetic level. For example, cerebellar damage is linked to circadian rhythm disruption, including sleep–wake phase disorders, through altered clock gene expression. Sgro and colleagues demonstrated sex differences in the expression of *Clock* and *rev-erb-α* genes in response to mild TBI in adolescent rats, where females showed reduced circadian time 0 h expression with sex-specific patterns for 24 h [120]. Furthermore, the knockout of the RBM5 gene in mice was associated with more visual deficits in males than females following TBI, which corresponded to the sexually dimorphic effects on the RIMS2 gene [121]. Understanding sex differences in disrupted gene expression in the cerebellum following TBI offers unique insight to functional recovery challenges and offers therapeutic targets to overcome them. This is important as cerebellar regions can be specifically tied to cognitive functions and disruption can induce neuropsychiatric dysfunction [115,119]. Collectively, these cerebellar differences, and their consequential impact on executive function and cognition, aid in understanding how the cortical atrophy of the cerebellum influences functional outcomes post-TBI.

## 3. Sexual Dimorphisms in Specific White and Gray Matter Compartments

White and gray matter tissues have different physical, metabolic, and cytoarchitectural properties which make them uniquely vulnerable to TBI effects. These unique variables influence the mechanical transmission of force and result in distinct pathophysiology and neurodegeneration. White matter is composed of axons that originate and terminate in gray matter, which can be myelinated or unmyelinated depending on developmental stage and specific function. Other white matter cells include oligodendrocytes, which function to myelinate the axons, and fibrous astrocytes, which serve a wide range of functions including modulation of inflammation [122,123]. White and gray matter compartments have specialized roles and effects on cognitive function, including processing speed, short- and long-term memory, and attention, as well as behavioral functions such as mood regulation. Damage to white and gray matter compartments caused by TBI are intrinsically related, as neurodegeneration of one compartment results in progressive loss of the other compartment via inflammation, excitotoxicity, and apoptotic signaling exchange [124,125,126]. Both white and gray matter have demonstrated sex-mediated differences, which in the context of TBI, could further affect tissue resiliency.

### 3.1. White Matter

White matter is a highly organized tissue with its own cytoarchitecture and developmental trajectories. White matter function is centered around communication conduction, whereby signals sent between gray matter regions are propagated by the myelinated axon tracts [127]. White matter integrity across complex network connections is integral, and damage is becoming increasingly associated with cognitive dysfunction and psychiatric disorders due to the chronic disruption of synaptic transmission and plasticity capability [127,128]. The interplay of sex, genetic, and environmental cues affect white matter development and contribute to increased vulnerability to dysfunction, as signals are no longer sent or received at their normal rate [127,129,130]. Patient age is also an important consideration when comparing studies due to rapid myelination changes during early childhood, corresponding to fractional anisotropy (FA) increases and medial diffusivity (MD) decreases [131]. Previous studies suggest sex differences in white matter are dependent on gonadal hormones and consequently appear during pubertal development [132,133]. However, Lawrence and colleagues found that sex influences in microstructure likely begin before the age of ten, with sex differences in oligodendrocyte-mediated myelination maturation [129,134]. Furthermore, as global white matter increases from childhood to adolescence, sex is seen to play a greater role in microstructural organization, as indicated by diffusion metrics [132].

Shearing forces sustained during TBI cause diffuse axonal injury which is correlated with white matter microstructure damage. Given the essential role of white matter in cortico-cortical communication dynamics, TBI disruptions induce long-term dysfunction. Growing evidence suggests that even mild TBI-induced white matter damage can result in chronic deficits in cognition and behavior [24,135]. This is especially concerning in pediatric patients as they are a high incidence demographic undergoing important neurodevelopmental processes. White matter microstructure damage, as evaluated by FA and diffusivity measures, predicts chronic recovery outcomes including mood disorders and social cognition and attention impairments [135,136,137,138]. Specifically, regional decreases in FA corresponded to 12-month post-TBI IQ scores [24]. Sex differences in FA measures localized to the uncinate fasciculus following TBI have demonstrated predictive value of time to symptom resolution, where Dennis et al. found the uncinate fasciculus, specifically, showed a greater change in association with more parent-reported behavioral problems in female pediatric patients [139,140].

Specifically, Novel Conduct Disorder and Oppositional Defiant Disorder were observed in approximately 14% of moderate-to-severe pediatric TBI patients with a significant association with frontal white matter lesions [141]. These behavioral disorders were further mediated by sex, as male children were more likely to develop comorbid ADHD and females had an increased predisposition to comorbid depression [141]. Nishat and colleagues found a correlation between disrupted superficial white matter maturation and increased internalizing behaviors, as well as increased deep white matter maturation rates and worse performance on the Picture Vocabulary Test and Pattern Comparison Processing Speed Test in female children following TBI [142,143]. Alternatively, in adult populations, males with sports-related concussions were found to have increased white matter impairment and reported worse symptoms compared to females [144]. Considering differences in myelination maturity between children and adults, this research indicates important age and sex interactions which influence TBI recovery.

Sex differences in white matter tracts connecting the orbitofrontal cortex and nucleus accumbens, which are implicated in reward and punishment behaviors, were linked to pubertal status in males, but not females, as males with increased fiber density reported less punishment sensitivity [145]. Similarly, a cross-sectional study found FA in the accumbofrontal tract peaked higher and earlier in males compared to females (13.9 vs. 18.6 years of age, respectively) [146]. These findings describe neural underpinnings which are associated with increased risk-taking behavior in males, and indeed converge with the results of a longitudinal study that found sensation-seeking relative to impulse control trajectories was higher, with longer windows of increased disparity, in males [147]. Given the involvement of the reward and punishment system, and its interplay with the cognitive control system, addiction and conduct disorders exhibit neural network correlations, thus lending further context to sex-influenced behavior outcomes post-TBI. Together, these studies indicate that sex differences in white matter microstructure differentially influences the progression of negative cognitive and behavior outcomes between males and females post-TBI.

### 3.2. Gray Matter

Total gray matter volume is generally accepted to decline after it peaks around 2 years of age in humans [148]. However, Gennatas et al. demonstrated that as gray matter volume decreases, gray matter density continues to increase, with strong age and sex associations [148]. For example, two brain regions may have the same gray matter volume, but one could have a higher gray matter density if the neurons are more tightly packed within that region. While females have lower total gray matter volume, their gray matter density is higher than that of males, such as the case of increased GABAergic neurons in the striatum [19,148]. Evidence indicates the variability of gray matter volume throughout the brain to be associated with sex from pediatric to geriatric demographics [149]. Gray matter symmetry, which is established during neurodevelopment, demonstrates sex influences in total gray matter volume and within specific neuroanatomical structures and regions [150,151]. A study spanning pediatric to adult data found that female brains were more symmetrical globally, with few structures and regions showing greater symmetry in males [150]. Comparatively, a pediatric study did not find a global sex effect on symmetry, but did report female children exhibited more regions of asymmetry, while male children showed a greater magnitude of asymmetry within specific regions [151]. In a large pediatric study, Kurth and colleagues noted significant increased asymmetry in the rostral anterior cingulate, thalamus, and nucleus accumbens in females, and significant increased asymmetry in the superior temporal gyrus and inferior parietal cortex in males, independent of age [151]. Whole brain and regional gray matter volumes are associated with cognitive ability and function, which are closely related to behavior [148,149]. Volumetric lateralization of total and regional gray matter, while commonly occurring under normal developmental conditions, was associated with predisposition to cortical diseases when incidence increased [150,152].

It is well established that TBI induces gray matter neurodegeneration in the acute window, which causes premature atrophy and predisposes patients to disease states associated with neurodegeneration, in addition to original injury-related functional deficits [53,67,153,154,155,156]. In adult cases, 80% of gray matter structures possessed higher atrophy trajectories, demonstrating brain age profiles that mismatched the patients’ chronological age due to progressive neurodegeneration post-TBI [153,156]. Similarly, children also demonstrated increased atrophy rates, which conflicts previously accepted hypotheses of children possessing greater inherent compensatory potential due to neural plasticity [154]. TBI sustained at an early age increases the risk of future disease due to the introduction of premature neurodegeneration during periods of critical growth and neurodevelopment. Because pre-injury gray matter content and sex-specific volumetric asymmetries determine the threshold of damage required to produce functional deficits, sex differences in total and regional gray matter critically shape recovery trajectories and must be accounted for when predicting long-term outcomes.

## 4. Sexual Dimorphisms in Functional Connectivity 

Functional brain networks undergo profound synaptic pruning and neural circuit changes during childhood and adolescence, thus leading to increased network integration, efficiency, organization, and specialization [157]. Unfortunately, in the context of TBI, these neural networks possess limited compensatory potential and therefore injury results in persistent neurological and functional deficits. Specifically, emerging evidence has found the default mode, dorsal attention, salience, social brain, and sensorimotor networks particularly implicated in pediatric TBI patients with notable differences between males and females (Figure 2) [158,159,160,161].

### 4.1. Resting-State Functional Connectivity

Resting-state functional connectivity occurs in the absence of an external stimulus and is related to cognitive, behavioral, and sensory functions [162]. Sex differences in functional connectivity have been demonstrated to emerge pre-puberty and persist well into adulthood [163]. For example, females aged 9–22 were found to exhibit more within-module connectivity in a convergent pattern across network scales, whereas males exhibited greater between-module connectivity [163]. The authors reported that individuals with more male-typical brain connectivity performed better in cognitive tasks that typically favor males, such as spatial accuracy, language reasoning accuracy, and speed in a finger tapping test of motor function [163]. Individuals with more female-typical brain connectivity performed better in cognitive tasks that typically favor females such as emotion identification and facial emotion recognition tests [163,164]. Supekar et al. supported these findings by demonstrating that female brain connectivity exhibited greater functional segregation and hierarchical organization of networks [165]. Collectively, these sex-specific differences in network organization and cognitive patterns explain why some individuals show greater resilience to functional deficits.

Another important functional connectivity consideration is lateralization (Figure 2). Agcaoglu et al. reported 6–10-year-old children exhibited both increases and decreases in lateralization of various networks at different stages of development, whereas adults only decreased with age [166]. Furthermore, males were found to have greater lateralization in activation of the left medial occipital gyrus, right middle frontal gyrus, right precentral gyrus, and right middle temporal gyrus components of the executive control network (i.e., cognitive control network) [166]. Females showed greater lateralization in the left superior parietal lobe of the sensorimotor network, and the right superior frontal gyrus of the default mode network [166]. The stronger lateralization of the executive control network in the right middle frontal gyrus in male children contributes to their higher prevalence of secondary ADHD after TBI [167]. Alternatively, female lateralization of the right superior frontal gyrus of the default mode network is an important factor for superior social cognition and self-awareness [168]. Contralaterally, increased resting state connectivity between the left superior frontal gyrus and the left fusiform gyrus seeds of the social brain network results in increased social dysfunction in adolescent TBI patients [160]. Because these regions participate in multiple networks that are susceptible to neural injury, they should be considered key mediators in social function post-TBI.

TBI disrupts brain functional connectivity through direct impact, diffuse injury, and secondary injury cascade variables, which damage functional nodes and connections. Resting-state connectivity disruption has been demonstrated to impact the social brain, default mode, sensorimotor, somatosensory, and executive control networks, in addition to others, which contributes to cognitive, behavioral, and motor dysfunction in both children and adults [158,159,160,169,170]. The social brain network, composed of connectivity between the superior temporal sulcus, temporal pole, and orbitofrontal cortex, among other anatomical regions, undergoes developmental maturation periods and is responsible for social cognition and behavior [160]. Alterations of this network following TBI have been linked to impairments in social functioning across age groups, but pediatric populations are likely to be predisposed to worse outcomes due to coinciding aberrant developmental disruption [160]. The default mode network plays a critical role in both acute and chronic outcomes following TBI. Preservation of default mode network integrity has been correlated with recovery of consciousness in the acute stage of TBI in adults, as well as improved response inhibition testing of behavior and cognitive function at chronic stages in children [159,171]. In pediatric TBI cases, increased connectivity between the default mode network and right dorsal premotor cortex correlated with worse motor coordination and control [158]. The dorsal attention network showed increased connectivity with the left sensorimotor cortex following TBI, which correlated with better motor performance, indicating a compensatory mechanism to restore motor function [158]. Pediatric TBI-induced increases in somatomotor–dorsal attention network connectivity are also associated with slower motor task completion [172]. The motor network is implicated in TBI outcomes as another neural underpinning for disruption of response inhibition, which is commonly seen in pediatric patients [173]. Together, these studies demonstrate network compensation can be either maladaptive or restorative. One common strategy for compensatory function is the recruitment of networks which can aid in top-down control of the damaged function.

Sexually influenced patterns of resting-state functional connectivity disruption following TBI include greater within-network connectivity in the motor, executive, and cerebellum networks in adult male TBI patients [174]. Conversely, more alterations in resting-state functional connectivity patterns between networks were found in female children compared to males, which persisted for at least a month after injury [161]. Networks which displayed sex-specific alterations in connectivity one month post-TBI included components of the salience and default mode networks, where females had more clusters of hypo- and hyper-connectivity than males [161]. Pediatric network dysfunction also does not show within-network variability, but does show decreased connectivity between visual and ventral attention networks, and between visual and default mode networks at acute and chronic timepoints post-TBI [175]. Furthermore, females with persistent symptoms had lower connectivity between dorsal and ventral attention networks than females without persistent symptoms, an effect not observed in males [175]. It is important to note that all three studies involved cohorts of mild TBI only, indicating sex differences in network connectivity are a sensitive phenomenon. Furthermore, Sharma and Wang et al. determined that challenges in delineating TBI symptom relationships with resting-state fMRI disturbance were better elucidated when considering sex as a biological variable [160,174]. Together, these data indicate network compensation strategies are profoundly different between males and females, regardless of age.

### 4.2. Task-Based Functional Connectivity

It is well documented that networks activated via external stimulation, referenced by task-based functional connectivity, have demonstrated age and sex effects [176,177,178,179]. For example, children have higher cerebral blood flow (CBF) in task-based networks which increases between 2 and 7 years old. However, males demonstrate a reduction in CBF until late puberty, which stabilizes to adult levels, and females demonstrate a mid- and late-puberty progressive increase, specifically in the executive and default mode networks [180,181,182]. Cellular mechanisms mediating this divergent response are linked to increased estrogen signaling during puberty and differences in cell signaling within the neurovascular unit [182,183]. Additional CBF studies have also found age and sex differences in task-based functional connectivity measures involving episodic and spatial memory tasks (Figure 2) [176,177]. The implication of sex differences in episodic memory tasks and memory-related disorders are correlated with age-related declines and differences in functional connectivity patterns of episodic encoding and the retrieval of spatial context processes [177]. Specifically, females showed greater activity in the lateral frontoparietal-related cognitive control and parahippocampal gyrus processes, which showed alterations in patterning as a function of advancing age [177]. Furthermore, age-related deficits in episodic memory are related to greater between-network integration in females and increased dorsal attention and default mode network connectivity in males [176]. Sex differences in the task-based functional connectivity of the sensorimotor network have also been observed, whereby females had lower brain activation and patterns of variability compared to males in motor tasks (Figure 2) [184].

In pediatric TBI, disruption of task-based functional connectivity, which results in impaired attention, can exacerbate functional deficits [178]. Pediatric patients who developed secondary ADHD following TBI experienced worse behavioral and cognitive recovery at a ~7-year follow up [185]. Attention and executive function deficits were observed 7 years post-injury, even in TBI patients without diagnosed secondary ADHD [186]. Attention-related deficits have been significantly associated with abnormal topology of gyri in the parietal and temporal regions, to the extent that a deep learning model was able to use alterations of these structures to identify pediatric TBI and predict future attention dysfunction [178,187]. Strazzer and colleagues reported that when performing a sustained attention task following brain injury, the altered activation of structures within the attention network led to a decreased ability to recruit additional components of the attention network as task difficulty increased [188]. This altered activation is indicative of a default compensatory strategy of the attention network. Similarly, increased activation in the frontotemporal regions in addition to other components of working memory circuitry have been associated with working memory impairments following pediatric TBI [178,179]. In the subacute period of TBI, Manelis and colleagues found that pediatric patients were unable to activate the left inferior frontal gyrus during more difficult working memory tasks and recruit the left orbitofrontal cortex 6 months post-injury [189]. Westfall and his team assessed 1-year post-TBI patients and found increased activation of the frontotemporal regions and more recruitment in comparison to healthy controls during difficult working memory tasks [179]. Similarly, motor task impairments were correlated with alterations in parietal functional connectivity, while decreased activation in the motor cortex and anterior cingulate were associated with impaired inhibitory control processing [178,190]. Impairments of the primary motor cortex, and to a greater extent the system-level functional connectivity of the left parietal cortex, were correlated with worse regional efficiency and inattention in children, which further correlated with decreased performance in attention, working memory, and motor function tasks following TBI [178]. Further research on sex-influenced connectivity patterns, particularly those involved in compensation and reorganization post-TBI, is essential for developing targeted therapies that improve functional recovery in pediatric patients.

## 5. Considerations and Limitations

The assessment of sex influences on pediatric neurodevelopment is challenging due to the complex interplay between other biological variables, variability in injury-specific parameters, and study limitations. Sex influences on TBI in both structural and functional vertices (Figure 1 and Figure 2) have a dynamic relationship with age and neurodevelopment. Furthermore, TBI patient databases have high variability in injury mechanisms, genetic backgrounds, and environmental cues, which serve as premorbid risk factors. Studies with larger sample sizes are necessary to further validate correlations and produce more robust conclusions.

Additionally, most of the research evaluated in this review focused on MRI-based analytical approaches, which can be quite heterogenous due to differences in MRI scanners, coils, collection protocols, post-processing pipelines, atlases, and other key analytics. Finally, a considerable limitation in assessing functional connectivity included discrepancies in structures involved within individual networks. For example, the superior frontal gyrus is considered part of the default mode network in some studies, but not all. This makes the comparison of networks across studies more difficult as the activation of a given structure can give rise to very different interpretations. The evaluation of sex differences in the context of pediatric TBI should therefore be characterized in the context of other biological variables and analytical techniques.

## 6. Future Directions

Sex is a key biological mediator of TBI pathophysiology as it shapes cellular responses, tissue vulnerability, and whole-brain structural and functional outcomes. Moreover, because TBI increases the risk of psychiatric and neurodegenerative disorders, sex-specific susceptibilities, including differing risks for ADHD, substance use disorders, mood disturbances, and cognitive impairments between male and female children, should guide future research in continuing to uncover underlying mechanistic and signaling pathways. These insights must also inform therapeutic development as pharmacologic agents often produce sex effects that influence efficacy readouts and translation to clinical trials. Furthermore, whether these pharmacologic strategies require sex-specific dosing or timing remains to be determined.

To advance precision medicine in pediatric TBI, sex-informed biomarkers and data standards should include MRI-derived measures of regional brain volumes, white and gray matter integrity, and functional connectivity. Treatment and rehabilitation plans should be tailored to sex-linked developmental trajectories, vulnerabilities, and cognitive–behavioral profiles. Integrating these approaches into diagnosis, treatment, follow-up, and family education programs will enhance recovery for pediatric TBI patients.

## 7. Conclusions

Pediatric TBI patients have been previously considered by the field to have a greater likelihood of recovery due to neuroplasticity. However, disruption of the dynamic, tightly controlled, ongoing development of the pediatric brain can instead result in worse outcomes across all injury severities. Patient-specific factors should be considered with the same degree of interest as injury-specific ones, but knowledge gaps in TBI pathophysiology continue to prevent development of treatments. Sex as a biological variable in pediatric TBI pathology is a critically understudied component of structural and functional neural connections of recovery. Sex influences in gray matter structural volumes and developmental trajectories may affect resiliency to threshold volume loss deficits and can induce premature atrophy in the developing brain. Gray matter compartmental volume and density and white matter microstructural integrity highlight innate sex effects which translate to greater internalizing behavioral deficits in female children, compared to greater externalizing behavioral deficits in male children. Sex influences in functional connectivity may further affect compensatory strategies following damage, and underlie cognitive, motor, and behavioral sequelae. Sex effects also show a strong interplay with age, illustrating nuanced differences in recovery throughout childhood and adolescence. Further elucidation of how sex influences neurodevelopment and functional recovery following pediatric TBI is an important step in addressing knowledge gaps to provide more informed care for this vulnerable patient population.

## Figures and Tables

**Figure 1 biomedicines-13-03033-f001:**
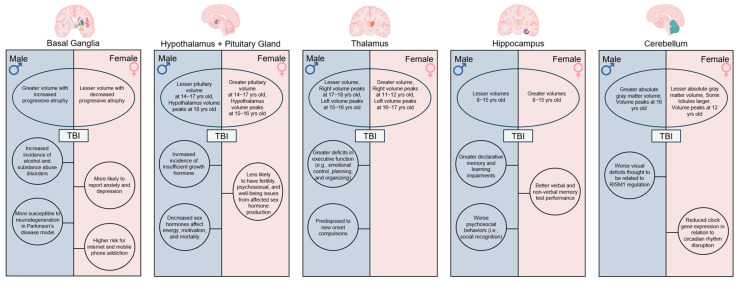
Sex Differences in neuroanatomy, neurodevelopment, and functional outcomes. Variations in anatomical volume and development of critical brain structures including the basal ganglia, hypothalamus, pituitary gland, thalamus, hippocampus, and cerebellum contribute to differences in TBI severity and recovery outcomes commonly observed between male and female children. Created in BioRender. Taber, M. (2025) https://BioRender.com/hlv5kpn (accessed on 11 July 2025).

**Figure 2 biomedicines-13-03033-f002:**
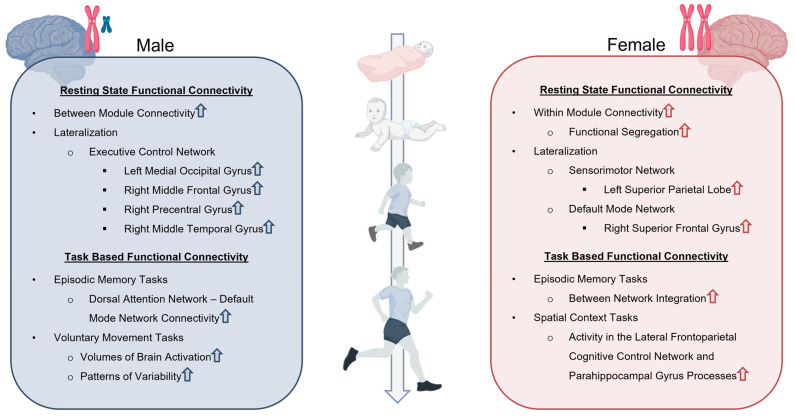
Sex Differences in resting state and task-based functional connectivity. Resting state and task-based functional connectivity analyses demonstrate unique differences in network organization, lateralization, and patterns of connectivity that are dependent upon both the age and sex of TBI patients. Upward arrows indicate increase in associated functional connectivity. Created in BioRender. Taber, M. (2025) https://BioRender.com/fi5rc7s (accessed on 11 July 2025).

## Data Availability

The analysis of this review article was pre-specified by the lead and corresponding author prior to the literature search. The resources and software used in this review are clearly identified and openly available. This article will be published under a Creative Commons Open Access license, and upon publication it will be freely available.

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
