# Peer review of "Sexual Dimorphisms in Neurodevelopment May Affect TBI Recovery in Pediatric Patients"

_biomedicines, 2025, doi:10.3390/biomedicines13123033_

Round 1

Reviewer 1 Report

Comments and Suggestions for Authors

Authors brought up an interesting and critical question: whether sexual difference among pediatric TBI patients would have impact on the function recovery. However, as authors mentioned several times in the manuscript, due to lack of sufficient data, it is hard to draw a definitive conclusion. Meanwhile, despite spending a lot of effort to describe the sexual dimorphisms in brain structure and functional connection under physiological and resting conditions that may eventually contribute to the different outcomes of pediatric TBI patients, authors didn't show strong and detailed data to support their hypothesis in context of trauma. Thus, find enough data or even wait more time collect enough data to first draw a clear conclusion about overall effect of sexual difference of pediatric TBI patients on functional outcomes becomes priority.

Author Response

Thank you so much for your time and thoughtful edits. Please see attached for responses. 

Reviewer 2 Report

Comments and Suggestions for Authors

It covers an important topic, however, it needs to elaborate more mechanistically  on the following topics in connection with cellular and molecular mechanism :

  • Pediatric Brain Development & Sex Differences

Gray/white matter trajectories, sexually dimorphic regions, functional network maturation.

  • Sex-Specific Responses to TBI

Structural and functional alterations post-injury.

Biomarkers, hormonal influences, cellular responses.

  • Functional Outcomes

Cognitive, behavioral, motor differences by sex.

How network disruption links to clinical outcomes.

  • Methodological Considerations

Sample size, female underrepresentation, sensitivity of imaging and analyses.

  • Knowledge Gaps & Future Directions

Longitudinal data, integration across scales, precision medicine. 

Cellular Mechanisms

  • Neurons

Sex-specific vulnerability to excitotoxicity, apoptosis, and axonal injury.

Differences in dendritic spine density and synaptic pruning during development.

  • Astrocytes

Sex differences in astrocyte reactivity and calcium signaling after TBI.

Influence on glutamate uptake and neuroinflammation.

  • Microglia

Differential activation patterns in males vs females post-TBI.

Role in synaptic pruning and neuroinflammatory cytokine release.

  • Oligodendrocytes / Myelination

How white matter maturation differs by sex and how TBI disrupts myelination trajectories.

  • Vascular & Blood-Brain Barrier (BBB) Changes

Sex differences in cerebrovascular reactivity and BBB permeability post-injury.

Microvascular rarefaction and perfusion deficits affecting recovery.

Author Response

(The authors gave the same response as above.)

Reviewer 3 Report

Comments and Suggestions for Authors

This review provides a comprehensive synthesis of current evidence on how sexual dimorphisms in neuroanatomy, neurodevelopment, white and gray matter, and functional connectivity influence recovery trajectories in pediatric traumatic brain injury (TBI). The authors cover major brain structures positioning sex as a critical yet understudied biological variable. They argue that sex-specific developmental trajectories intersect with TBI-related damage to influence cognitive, behavioral, motor, and psychosocial outcomes. The review highlights substantial gaps in existing research, including methodological inconsistencies, small sample sizes, and limited longitudinal and female-specific data.

Comments

several weaknesses limit the study impact.

Conceptually, The manuscript is highly descriptive in multiple sections, leading to redundancy and a narrative that at times lacks a clear critical stance toward conflicting or insufficient evidence. Many claims of sex-specific differences rely on sparse or inconsistent pediatric data, with some conclusions extrapolated from adult studies without fully acknowledging developmental mismatches. Additionally, the manuscript emphasizes volumetric and connectivity differences but offers limited integration of mechanistic pathways or unified models explaining how sex interacts with neurodevelopment to shape TBI outcomes.

Methodologically:

Methodological heterogeneity—such as varying MRI pipelines, small sample sizes, inconsistent network definitions, and underrepresentation of females—is acknowledged but not critically evaluated in terms of how these limitations affect data interpretation.

Discussion: 

The discussion offers limited translation to clinical practice, missing opportunities to propose tangible implications for assessment strategies, rehabilitation planning, biomarker development, or precision medicine pathways.

The authors should also expand the clinical implications, offering more explicit guidance on how sex-informed insights may influence prognosis, treatment, or long-term follow-up in pediatric TBI. Please assessthe above comments to strengthen the different sections highlighted.

Author Response

(The authors gave the same response as above.)

Round 2

Reviewer 1 Report

Comments and Suggestions for Authors

Appreciate authors' efforts to address questions and concerns one by one in a responsible and logical manner. After revising, the conclusion becomes more sound. 

Reviewer 3 Report

Comments and Suggestions for Authors

 Accept in present form